# Understanding children's experiences of self-wetting in humanitarian contexts: An evaluation of the Story Book methodology

**Claire Rosato-Scott**[1], **Mahbub-Ul Alam**[1,2], **Barbara E. Evans**[1], **Joanne Rose**[3], **Eleanor Wozei**[4], **Dani J. Barrington**[1,5]*

**1** School of Civil Engineering, University of Leeds, Leeds, United Kingdom, **2** Infectious Disease Division, International Centre for Diarrhoeal Disease Research Bangladesh (icddr,b), Dhaka, Bangladesh, **3** Department of Health Science, University of York, York, United Kingdom, **4** Department of Engineering and Environment; Ugandan Christian University, Mukono, Uganda, **5** School of Population and Global Health, University of Western Australia, Crawley, Australia

* dani.barrington@uwa.edu.au

**Data Availability Statement:** There are ethical restrictions imposed by the Research Ethics Committee, Faculty of Engineering, University of

## Abstract

Little is known about how children in humanitarian contexts experience self-wetting. Children can wet themselves due to having the medical condition of urinary incontinence (the involuntary leakage of urine), or due to them not wanting to or not being able to use the toilet facilities available (social or functional incontinence). Self-wetting is a global public health challenge: the physical health of children can suffer; they can miss out on educational and social opportunities; they may face increased protection risks; and the emotional effect on daily life can be significantly negative. The Story Book methodology was developed to facilitate conversations with children aged five to eleven in humanitarian contexts (specifically refugee settlements in Adjumani District, Uganda; and refugee camps in Cox's Bazar, Bangladesh) about self-wetting to understand how humanitarian professionals can best meet the needs of children that wet themselves. This paper has evaluated how far the Story Book methodology meets the specific requirements of conducting research a) in a humanitarian context; b) with young children; and c) on a personal and highly sensitive topic. Data has been used from Story Book sessions held with children in Adjumani District and Cox's Bazar, and from semi-structured interviews held with adults known to have participated in the planning and/or facilitation of the sessions. The evaluation found that although the Story Book methodology provided deep insights into how children in humanitarian contexts experience self-wetting, it was not always implemented as designed; it is not practical to implement in humanitarian settings; and it was not acceptable to all participants and facilitators as a research tool. Changes have been recommended to improve the methodology as a research tool to better understand how children experience personal health issues, but even with such changes the methodology will remain better suited to non-humanitarian contexts.

Leeds, United Kingdom which prevent the public sharing of sensitive minimal data for this study. Data are available upon request from the Research Ethics Committee, Faculty of Engineering, University of Leeds, United Kingdom via email (MEECResearchEthics@leeds.ac.uk) for researchers who meet the criteria for access to confidential data.

**Funding:** 'Understanding children and their caregivers' experiences with incontinence in humanitarian contexts' (Project #45432, Principal Investigator Dr Dani J Barrington) was funded and supported by Elhra's Humanitarian Innovation Fund (HIF) programme, a grant-making facility which improves outcomes for people affected by humanitarian crises by identifying, nurturing, and sharing more effective, innovative, and scalable solutions. Elhra's HIF is funded by aid from the Netherlands Ministry of Foreign Affairs. The funders had no role in study design, data collection and analysis, decision to publish, or preparation of the manuscript.

**Competing interests:** The authors declare only the following which may be considered as a potential non-financial competing interest: Mahbub-Ul Alam is currently serving on the editorial board of PLOS Global Public Health. This does not alter our adherence to PLOS policies on sharing data and materials.

# Introduction

## Background

The Story Book methodology has been developed to facilitate conversations with children aged five to eleven in humanitarian contexts (specifically refugee settlements in Adjumani District, Uganda; and refugee camps in Cox's Bazar, Bangladesh), about a personal and sensitive health issue: self-wetting. Self-wetting is a global public health challenge due to the far-reaching consequences experienced by children who self-wet, yet the experiences of self-wetting children in humanitarian settings have not been explored. The Story Book methodology has therefore been designed to meet the specific requirements of conducting research a) in a humanitarian context; b) with young children; and c) on a highly sensitive topic. This paper is an evaluation of how far the Story Book methodology meets these requirements and can therefore be considered as a research tool for prompting discussions around self-wetting (and sensitive issues more broadly) to support evidence-informed humanitarian programming.

## Children's experiences of self-wetting

Children aged between five and eleven years old sometimes wet themselves. This could be due to them having the medical condition of urinary incontinence (UI) defined as the involuntary leakage of urine. Or due to them not wanting to use, or not being able to use, the toilet facilities available (known as social, or functional incontinence).

No matter why a child wets themselves, the consequences are the same. Physical health can suffer; they may experience rashes, urinary tract infections and even dehydration if they restrict liquid intake to reduce the need to urinate [1]. They can miss out on educational and social opportunities; they may have increased protection risks due to caregiver frustrations in the home and/or the stigma of incontinence in the community; and the emotional effect of the condition on daily life can be significantly negative [2]. Self-wetting is therefore a global public health challenge.

## Displaced children's experiences of self-wetting are unknown

The United Nations International Emergency Children's Fund (UNICEF) estimates that 36.5 million children were displaced from their homes at the end of 2021, due to conflict, violence and other crises [3]. Little is known about how displaced children understand and experience self-wetting, and how they could be better supported. Spencer et al. [4] found that there is a "distinct paucity of (migrant) research [on health] that takes children's perspectives as its starting point" and, to the best of the authors' knowledge, children in humanitarian contexts have not been spoken to about self-wetting. As a result, humanitarian programmes in sectors including health, protection, and water, sanitation and hygiene (WASH) may not meet the needs of children that wet themselves.

## The challenges of conducting research on self-wetting with displaced children

It is difficult to conduct research in a humanitarian setting. Such contexts–at least in the initial stages–are characterised by disruption and instability which present unique challenges to researchers. These include securing adequate resources (financial, technical, human and time), difficulty in accessing study populations, interruptions to public services, and the inappropriateness of traditional research methods [5, 6].

Researchers are beginning to share field experiences on how research has been conducted in humanitarian settings [notably 7] and strategies are emerging to address the challenges

faced in such a context. These include using flexible, adaptive and iterative methodologies that produce quick, real-time data; collaboration between academic institutions (experienced in research design and analysis) and humanitarian organisations (with established local relationships, to help address logistical and security challenges, and to ensure that research findings will benefit the affected populations) [7–9]. Finally and most crucially is the engagement with affected populations to enable trust, improve research design and facilitate the dissemination of findings [7, 8].

Conducting research with children in a humanitarian setting presents further challenges. Children are rights-holding individuals and have a right to be heard including in 'situations of crisis or in its aftermath' [10, 11]. Participation has emerged as a concept to describe efforts to implement a child's right to be heard, and it has been shown that children benefit from being involved in matters that concern them (by, for example, contributing to rehabilitation and strengthening a sense of identity), albeit "care needs to be taken to ensure that participation does not result in exposure to traumatic or harmful situations" [11 Paragraph 125,12].

Professionals in humanitarian contexts aware of the particular vulnerabilities of displaced children may consequently–and understandably–emphasise the protection of children and limit their participation in the design of humanitarian programmes. Guidance has therefore emerged to encourage the inclusion of children in research conducted in humanitarian contexts by outlining ethical principles to adhere to, including beneficence (promote well-being), non-maleficence (do no harm) and justice (consider who is burdened and who benefits) [13–16].

Finally for consideration are the unique requirements of conducting research on personal and highly sensitive health issues. All known research conducted with adults (none known has been conducted with children) on self-wetting in low- and middle-income contexts has found that there is such a stigma associated with the condition that it is rarely spoken about by those experiencing self-wetting, their caregivers, or the wider community [17–20]. It is therefore a topic that is 'best-suited to in-depth discussions, which are flexible in structure and guided by the participant' with measures taken to reduce any embarrassment or discomfort, for example, by using simple language; having discussions in comfortable, private locations; and having men speak with men and women speak with women [21].

## The Story Book methodology

In 2019, a Research Team (RT) from University of Leeds (United Kingdom, UK), The University of Western Australia, University of York (UK), Plan International Uganda (PIU), Plan International UK, Uganda Christian University (UCU), UNICEF Bangladesh and World Vision Bangladesh (WVB) was awarded funding from Elhra's Humanitarian Innovation Fund (HIF) for the research project 'Understanding children and their caregivers' experiences with incontinence in humanitarian contexts'.

As–to the best of the authors' knowledge–children in humanitarian contexts had not been spoken to about self-wetting before, the RT, in consultation with an Advisory Board consisting of experts in humanitarian affairs, incontinence and conducting research with children, developed the Story Book methodology. The methodology was developed to facilitate conversations with children aged five to eleven in refugee settlements in Adjumani District, Uganda and refugee camps in Cox's Bazar, Bangladesh about their experiences of this particular health issue. The methodology is a drawing-based research method used in a small group (up to 6 children) setting led by a facilitator. It is designed to address the operational challenges of conducting research in a humanitarian setting (for example, it is implemented through collaboration between academic institutions and humanitarian organisations); the ethical challenges of conducting research with children (for example, by holding discussions within a group context

which provides peer support rather than using an interview format); and the specific challenges of conducting research on self-wetting (for example, no participant was asked to share any personal experiences, sharing instead the views of a 'hero' character (Table 1)). For more details on how the methodology was developed see S1 File and [16]; to view the original study tools see [22].

**Table 1. Summary of the session guide in which the Story Book methodology was implemented.**

| Agenda item | Detail | Minutes |
|---|---|---|
| Greetings and introduction to the session | The facilitator(s) and observer(s) introduce themselves. The facilitator explains the purpose and structure of the session. The facilitator recaps key assent-related messages including that participants can leave at any time without penalty. Purpose: To ensure that all participants have assented to participation. | 5 |
| Ice-breaker | The facilitator chooses an appropriate ice-breaker. Purpose: The ice breaker activity makes children feel more comfortable with each other and with the subject matter of the focus group. | 10 |
| Activity 1: Co-creating a hero | The facilitator supports the group to create a main character, or 'hero', for their Story Book. The children are asked to choose a gender, age, name, who the hero lives with, the hero's favourite animal, etc. The facilitator draws the hero as guided by the children. Purpose: As a group, the children create the 'hero' for their book. Note that the use of an *imaginary hero* rather than asking participants to share personal experiences of self-wetting reduces the risk of a) a child becoming distressed at being asked to share such experiences and b) a participant being identified by friends, family and the wider community as experiencing self-wetting which may result in negative consequences due to the stigma associated with the condition. | 5 |
| Activity 2: Introducing the idea of self-wetting | The children are asked to draw the hero doing their favourite activity, for example playing football, and how they feel doing it. The facilitator then explains that the hero has just wet themselves and asks the children to draw how the hero now feels. At this point the group may discuss reasons why the hero wet themselves. Purpose: As a group, the children begin to explore the feelings of the hero, including when hero has self-wet. | 25 |
| Extended snack and play break | To refresh energy levels. | As needed |
| Activity 3: Exploring a day in the life of the hero | The children are asked to describe what they do as soon as they wake-up. The facilitator then explains that the hero has woken up to find that they have wet the bed and asks the children to draw how the hero feels. The children are then asked to draw how the hero's caregiver reacts to the hero wetting the bed. The facilitator then explains that the hero has now gone to school and wets themselves there. The children are asked to draw how the hero feels. The children are then asked to draw how the hero's teacher reacts to the hero wetting themselves at school. At this point the group may discuss ideas for improving the day of the hero. Purpose: As a group, the children continue to explore the hero's feelings and experiences related to self-wetting, and the reactions of friends and community members to better understand the consequences of self-wetting and any stigma associated with self-wetting. | 40 |
| Close | The facilitator thanks the children for taking part and explains how their drawings will be used. | 5 |

The RT provided training on the methodology to facilitators in both locations, using a proposed agenda for a 90-minute session (excluding break) (Table 1):

## Methods

### Summary

In each location, households with potential participants were identified and consent (from adult caregivers)/assent (from child participants) was sought; to view the forms see [22]. Adult caregivers provided written consent where possible. When unable to provide written consent (for example when illiterate), two data collectors witnessed the caregiver's verbal consent on a written consent form. Children provided verbal assent only. Sessions using the Story Book methodology were held in Cox's Bazar in October 2021, and in Adjumani District in February 2022. To evaluate the Story Book methodology, outputs from the session were analysed and semi-structured interviews with adults known to have participated in the planning and/or facilitation of the sessions were also conducted.

### Story Book session: Data collection

In Adjumani District, the sessions were conducted by PIU and UCU in February 2022. Adjumani District is located in northern Uganda and hosts a number of refugee settlements. The majority of refugees are from South Sudan, fleeing a civil war which began in December 2013. Uganda has progressive policies towards refugees relative to neighbouring countries, and refugees have the right to work, the right to the same social services as host communities (health and education for example) and freedom of movement [23]. Settlements are therefore long-established and as at 31 January 2022, there were over 244,000 refugees residing in Adjumani District of which 39,000 (16%) were children aged five to eleven [24].

In Cox's Bazar the sessions were conducted by WVB in October 2021. Cox's Bazar is located on the south-eastern coast of Bangladesh, and it is the world's largest refugee settlement. The majority of refugees in Cox's Bazar belong to the Rohingya people, a Muslim ethnic-minority group who have lived for centuries in Myanmar. The Rohingya are not officially recognised as an ethnic group in Myanmar and have faced decades of persecution, forcing many to flee to neighbouring countries including Bangladesh, India and Thailand. In August 2017, violence in Myanmar's Rakhine State triggered the largest and fastest evacuation: as at 28 February 2022, there were 923,179 Rohingya refugees living in Bangladesh across 34 camps within Cox's Bazar District and on the island of Bhasan Char, of which over 200,000 (22%) were children aged between five and 11 years old [25].

In total, nine sessions were conducted in Adjumani District and eight in Cox's Bazar (Table 2). In Bangladesh the focus group facilitators were hygiene officers used to working with children and known to participants through their ongoing presence in the camp. In Uganda the facilitators were research assistants from the Plan International Uganda database, known to have experience in qualitative data collection and who were familiar with the local community. All facilitators spoke the local languages of the child participants. Sessions were conducted in locations familiar to the children, for example child-friendly spaces, and caregivers were not present (although in Uganda caregivers observed the focus groups from a distance). Children were assigned to focus groups based on their gender, age (5 to 7 years or 8 to 11 years) and residential location only, and had similar socioeconomic and educational status. Participants in a focus group may or may not have known each other prior to taking part in the focus group. Further demographic data on the participants can be found in Table A in S2 File.

**Table 2. Story Book session reference by location.** See Table A in S2 File for additional data on sessions held in Cox's Bazar.

| Session | Adjumani District | | | Cox's Bazar | |
|---|---|---|---|---|---|
| (Gender / age in years) | Ayillo | Pagirinya | Pagirinya host community | Camp 7 | Camp 8E |
| Boys / 5 to 7 | AD1 | AD4 | AD8 | CB7 | CB1 |
| Girls / 5 to 7 | AD2 | AD5 | AD9 | CB8 | CB3 |
| Boys / 8 to 11 | AD3 | AD6 | | CB5 | CB4 |
| Girls / 8 to 11 | | AD7 | | CB6 | CB2 |
| **Total by location** | **3** | **4** | **2** | **4** | **4** |
| **Total sessions** | **17** | | | | |

## Evaluation of the Story Book methodology

Outputs from all sessions were made available to the RT for analysis via approved and secure platforms and uploaded to NVivo 12 for analysis by the lead author. The outputs analysed were de-identified, translated transcripts of the facilitators describing the drawings from the sessions held in Adjumani District; de-identified, translated transcripts of the sessions held in Cox's Bazar (these were not available in Adjumani District as the children preferred to draw answers only); photographs of the drawings from all sessions; and facilitator field notes.

The lead author also conducted semi-structured interviews with five adults known to have participated in the planning and/or facilitation of the sessions in Adjumani District or Cox's Bazar (referred to as ADI1; ADI2; ADI3; CBI1; and CBI2). The interviews were conducted in early-2022 remotely using teleconferencing software. Verbal informed consent was obtained from each participant before beginning the interview. De-identified transcripts were stored on an approved and secure platform and uploaded to NVivo 12 for analysis by the lead author.

Coding (Table 3) was undertaken using NVivo 12 to analyse the transcripts (of both the Story Book sessions and semi-structured interviews); drawings; and facilitator fieldnotes. The data was first deductively coded using Bowen et al.'s (26 p.3) proposed areas of focus for a feasibility study, considering that the Story Book methodology was designed to meet the specific requirements of conducting research a) in a humanitarian context; b) with young children; and c) on a highly sensitive topic. This analysis framework was selected as it is designed to determine whether a new intervention–such as the Story Book methodology–is appropriate for further testing and if any modifications are needed prior to its further use, as a means to identify those interventions which should be advanced for further testing as they have a high

**Table 3. Areas of focus adapted from Bowen et al.'s [26] p.3].**

| Area of focus | Definition |
|---|---|
| Implementation | How the Story Book methodology was implemented in practice versus how it was designed to be implemented (given that it was designed to meet the requirements of a) conducting research in a humanitarian context, b) with young children, on c) a highly sensitive topic) |
| Practicality | The practical impact of resources (financial, technical, human and time) on the implementation of the Story Book methodology (and therefore how does the methodology meet the requirements of a) conducting research in a humanitarian setting) |
| Acceptability | How acceptable the child participants found the Story Book methodology (and therefore how does the methodology meet the requirements of conducting research b) with young children on c) a highly sensitive topic) |
| Adaptation | Modifications made to the Story Book methodology due to context (and therefore how does the methodology adapt to use in a) humanitarian contexts, on c) highly sensitive topics |

Note that Demand, Expansion, Integration, and Limited-efficacy testing were not deemed to be appropriate areas of focus for this study.

probability of efficacy [26]. Inductive coding was then used within each area of focus to identify themes, and verbal answers given by the children during the sessions in Cox's Bazar were also coded.

## Ethical considerations

In Adjumani District, households were purposively selected by data collectors familiar with the local communities, being households with children aged five to eleven years old known to experience self-wetting as reported by caregivers. The RT within Uganda (PIU and UCU staff) believed this was the most appropriate way of collecting information on children and caregivers' experiences of self-wetting in this context). In Cox's Bazar, data collectors familiar with the local communities used inclusion criteria–gender (boys and girls); age (five to eleven); living with an adult caregiver–to identify households with potential participants. Purposive selection criteria was not used to identify participants known to experience self-wetting, or be more likely to experience self-wetting (for example, children with a disability), as the RT in Bangladesh (author MUA and WVB staff) felt that the risks of causing personal distress to participants during the Story Book session and/or negative consequences should a child with self-wetting be identified by the wider community, were too high in this context. The RT in Bangladesh concluded that gaining an understanding of the general awareness about and attitudes towards self-wetting, rather than direct personal experiences, was felt to be sufficient to achieve the research objectives whilst protecting the participants from undue harm.

Approval to conduct the project 'Understanding children and their caregivers' experiences with incontinence in humanitarian contexts', including the lead author's evaluation of the Story Book methodology, was granted by the Research Ethics Committee, Faculty of Engineering, University of Leeds, United Kingdom (Reference MEEC 19–020). Approval to conduct the research in Cox's Bazar was granted by the Institutional Review Board of the Institute of Health Economics (University of Dhaka, Bangladesh), with authority to access the refugee camps granted by the Office of the Refugee Relief and Repatriation Commissioner. Approval to conduct the research in Adjumani District was granted by the UCU Research Ethics Committee (Reference 2021–82) and the Uganda National Council for Science and Technology, with authority to access the refugee settlements granted by The Prime Minister's Office of the Uganda Government.

## Inclusivity in global research

Additional information regarding the ethical, cultural, and scientific considerations specific to inclusivity in global research is included in S3 File.

## Results

### Implementation

The Story Book methodology was designed to meet the requirements of a) conducting research in a humanitarian context, b) with young children, on c) a highly sensitive topic). However, it was not always implemented as designed (not all activities always took place, most sessions ran over the intended maximum time of 90 minutes, and participants became tired and lost concentration at times), and the data collectors struggled to interpret the data. It therefore did not meet the requirements of conducting research in a humanitarian context (the sessions were too time-intensive and actions to be taken were unclear), and with young children (participants lost interest at times).

The overriding concern of the RT when designing activities for the Story Book session was that the child participants should not experience distress, with measures taken including that they were not to be asked to share personal experiences of self-wetting. In Adjumani District, the children spoke very little during the sessions as '*it was not an easy subject to handle, for the children to open up*' (ADI2) so '*those [who] could talk at all, were talking very, very quietly. We could see that it was a struggle to just make them talk because of the experiences I think each one of them has gone through*' (ADI1). However, '*you could really see that [the children] are very relaxed*' (ADI2). Facilitator field notes in Cox's Bazar reported that '*the children feel free to discuss their incontinence through their drawings as they express it anonymously*' although there was one instance of a group being asked to share personal experiences of self-wetting: 'Facilitator: *Let me tell you about myself; I felt humiliated . . . How did you feel then*? Child 1: *I used to feel ashamed*' (CB5). Despite reports that the children were '*happy because for them [the Story Book session] was fun to do*' (CBI1), one interviewee on reflection was still concerned that children should not be spoken to about personal health issues as '*it (puts) a kind of fear in them*' (CBI2).

The Story Book session was designed to have three activities (Table 1): co-create a hero (Activity 1); introduce the idea of self-wetting (Activity 2); and explore a day in the life of the hero (Activity 3). In Adjumani District workbooks were used with an allocated page for each activity that included a pre-drawn prompt (Activity 1: the outline of a person and a home; Activity 2: a blank face; Activity 3: a rising sun, a sun, and a moon; see 'Drawing Sheets for Story Book' at (22); and all activities were completed by all FGDs. In Cox's Bazar, all sessions began with Activity 1, but only two (CB6 and CB8) did Activity 2 as designed with all other sessions moving straight to Activity 3 (which eventually CB6 and CB8 did too). The Story Book session guide (Table 1) also suggested that the groups discuss why the hero may self-wet, and ideas for how to improve the day of the hero. These conversations were not held in Adjumani District given the reluctance of the children to speak during the sessions, but it was suggested that '*there . . . could be two or three questions given to the children to express verbally, maybe at the end of the drawing, they would be in position to talk [as] you could see that they were more willing to open up on the subject*' (ADI2). In Cox's Bazar discussions were held in most groups (CB3 and CB4 did not discuss why the hero may self-wet).

The Story Book sessions were designed to be 90 minutes in length, excluding a break between Activities 2 and 3 (Table 1). In Adjumani District the sessions (including breaks) took between 60 and 120 minutes (exact times are not known as the sessions were not audio recorded). Despite concerns noted in some of the transcripts–'*[Participant 3] also has the same story with [Participant 2] and I suspect they did not have enough time*' (AD1)–those interviewed felt that the sessions were about the right length of time. Story Book sessions in Cox's Bazar (excluding breaks) ranged from 50 minutes (CB3) to 155 minutes (CB5), with an average length of 116 minutes (Table A in S2 File). Interviewees confirmed that a shorter session, as designed, would have been preferred: '*(the session) should be within one hour or one hour 30 minutes for both ages . . . (with a) break after every 30 minutes*' (CBI1).

## Practicality

The Story Book methodology was designed to meet the requirements of conducting research in a humanitarian context, but it was not found to be practical to implement in humanitarian settings because it is too resource-intensive (in particular, human and time).

Interviewees in Adjumani District raised that '*it would have been much easier if we had been there maybe for a week or two*' (ADI2) for the community to feel more comfortable with the researchers, which may have encouraged more verbal discussions during the FGDs. Facilitator

field notes stated that '*the activity was very tiresome*' but ADI1 commented that '*another thing that worked out very well, those refreshments to make them relaxed*' and '*in fact I think the part where . . . the children are made to be as relaxed as possible, I think that [should] be emphasised more*'. In Cox's Bazar the children became tired and lost concentration at times: '*Do the exercise; they have lost their concentration*' (Facilitator, CB5); '*The concentration is lost, they laugh*' (Facilitator, CB7). The facilitators did use physical activities such as dancing and snacks to revive energy levels–'*Ok, are you feeling tired? Let's stand if we feel tired, dance a little*' (Facilitator, CB2)–but despite such efforts '*going to the end, [the children] feel bored about . . . doing . . . more drawings about anything*' (CBI2). This impacted participation: '*after a certain period of time, the children has (sic) become so tired and they don't want to do (sic) participate anymore, especially after activity two*' (CBI1).

## Acceptability

The Story Book methodology was designed to meet the requirements of conducting research with young children on a highly sensitive topic, but it was not always found to be an acceptable method for the facilitators and participants, and particularly for children aged 5 to 7.

An interviewee in Adjumani District reported that '*with the methodology, you could see that . . . despite the circumstances, the drawing and the Story Book, helped [the children] to open up much more easily than maybe you would have expected*' and felt that '*if we're doing real conversation, if we're talking to the children, I think it would have been very complicated . . . I don't think the children would have [been] happy to talk to us on that subject*' (ADI2). As the '*child was asked to imagine it is somebody else made it feel comfortable for the children to respond*' so much so that by '*the end of [the session] you could see that [the children] were relaxed and they felt that they had some someone who could talk to them and understand that was not cruel and the subject was not an issue of conflict*' (ADI2); '*the children they felt they were being considered*' (ADI3). However, despite such positivity it was noted that if purposeful selection criteria had not been used in Adjumani District (and it was not in Cox's Bazar), participants would '*have been more relaxed and more open*' (ADI2).

Issues with the activities were noted in Adjumani District, with facilitators noting that '*the baby ones [the 5 to 7 years olds] were unable to shade and even draw the character of the story*' (AD Facilitator field notes) and '*[the 5 to 7 year old girls] need a lot of patience and guidance on what to do, otherwise they will just colour in a free style*' (AD2). Similar issues with the activities were noted in Cox's Bazar with some of the children needing support to think of an answer to the questions being asked: '*the eight to 11 years old children are quick[er] to grab the question . . . than the age group of five to seven years old children because it is tough to make them understand the activity*' (CBI1). This may have contributed to the facilitator suggesting answers: '*Facilitator 1: She couldn't control her urine, so isn't she feeling bad? Why is she feeling bad? Facilitator 2: Isn't she annoyed because she has no control over herself?*' (CB6). Of the 72 answers given to describe how the hero was feeling after self-wetting (at play, home and school), 13 (or 18%) were initially suggested by the facilitator (Table B in S2 File). Of the 62 answers given to describe how friends, caregivers and teachers reacted to the hero after self-wetting, 7 (or 11%) were initially suggested by the facilitator (Table C in S2 File).

Even once a question was understood and a response decided on, some found the concept of drawing emotions too abstract: '*How do I draw these emotions?*' (Facilitator, CB5); '*I can't draw like that*' (Child, CB5). This resulted in facilitators drawing on behalf of the children at times: '*the early ages children . . . we have to nudge them, we have to say . . . You can say to me, I can draw for you. And we did that actually*' (CBI2). There was also confusion about what the children were drawing at times: '*[Participant 3] I think she was drawing a bed, it is quite strange*

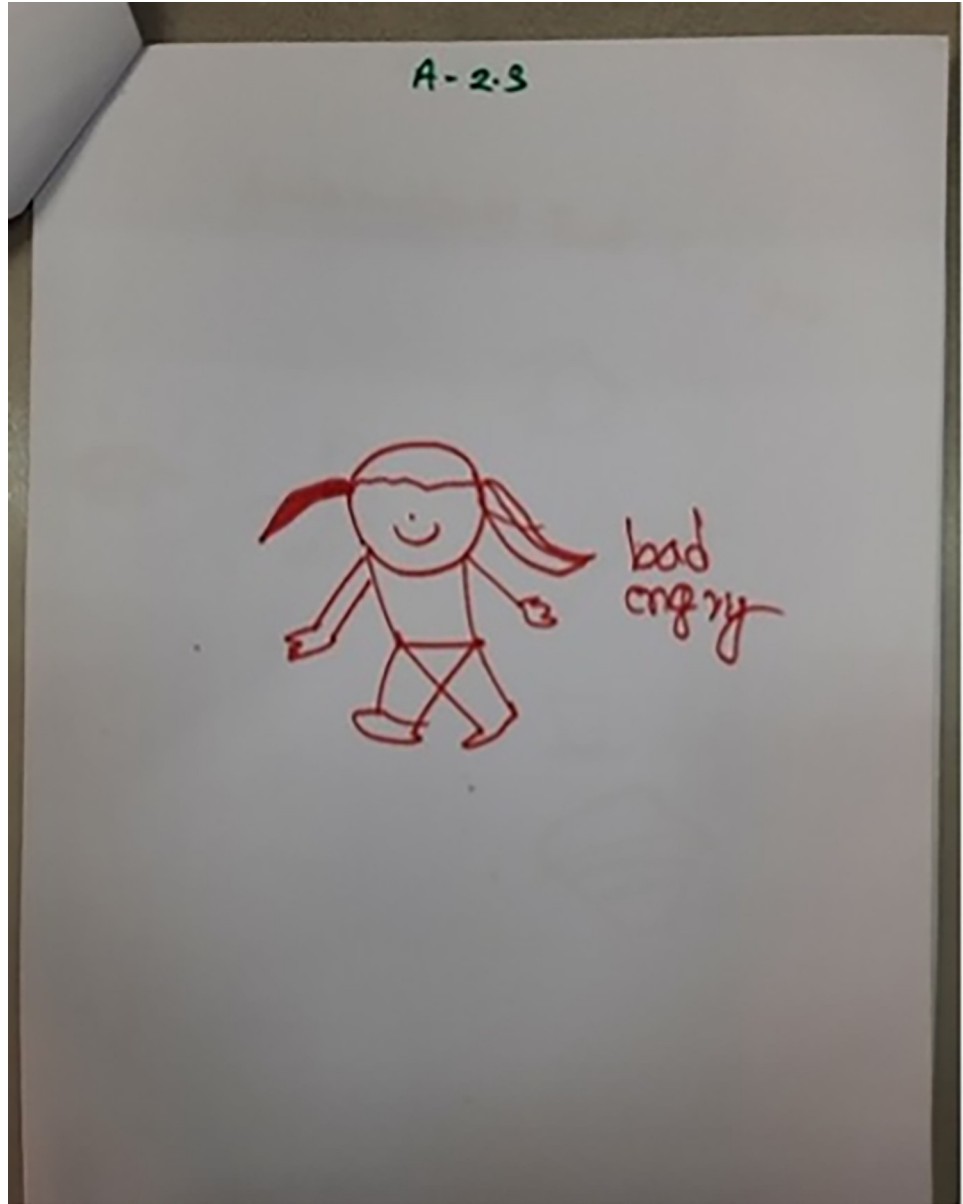

**Fig 1. 'Bad angry' (CB4).** Adding descriptors to explain drawings (Photographs by Sudipta Das Gupta). Note that the descriptions of the emotion do not clearly match the faces of the drawn figures.

*and not easy to define'* (AD2); '*he is saying one thing but has drawn another'* (Facilitator, CB1). In Cox's Bazar the names of the emotions that the children were attempting to draw were therefore written onto some of the drawings by either the facilitator or child (Figs 1–4).

So, whilst '*drawing is the perfect way to understand the incontinence issue of the children, because the children are not comfortable to say this incontinence issue, even to their parents . . . drawing human figure is tough for them*' (CBI1). It was proposed that '*more pictures should appear so that it will be easy for [the children] to give answers'* (AD Facilitator field notes) and/ or '*emojis could be a better options'* (CBI2) which would also '*reduce the time'* (CBI1) of the FGDs. It was also suggested that a psychologist would have been better placed to facilitate the

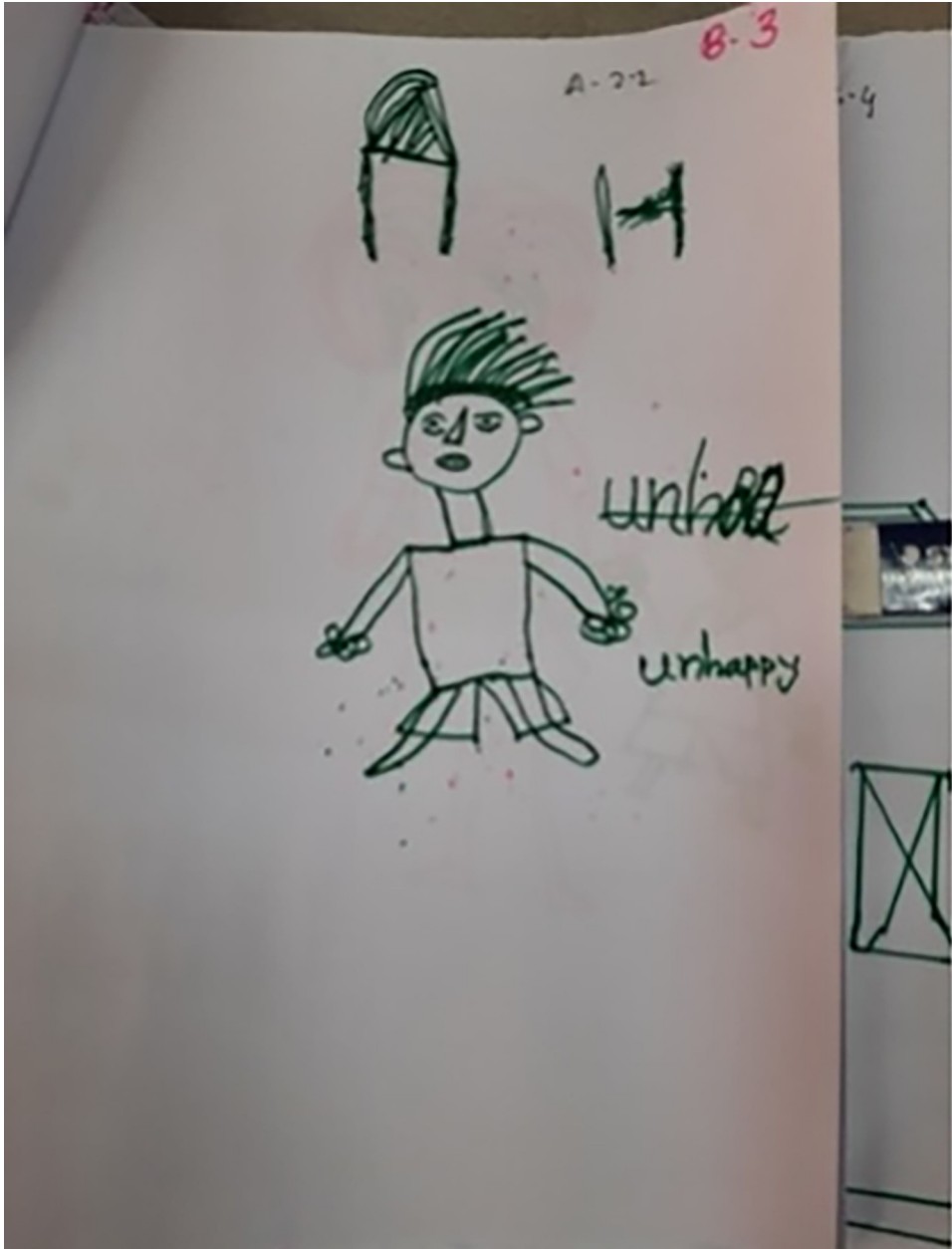

**Fig 2. 'Unhappy' (CB5).** Adding descriptors to explain drawings (Photographs by Sudipta Das Gupta). Note that the descriptions of the emotion do not clearly match the faces of the drawn figures.

FGDs and interpret the drawings: '*we are trying to read their mind but instead of me if there are any psychologist . . . then they could do it . . . more clearly*' (CBI2).

## Adaptation

The Story Book methodology was designed to be adaptable for use in humanitarian contexts and on highly sensitive topics, but the design was time-intensive and it was not found to have been sufficiently adapted for use in Adjumani District.

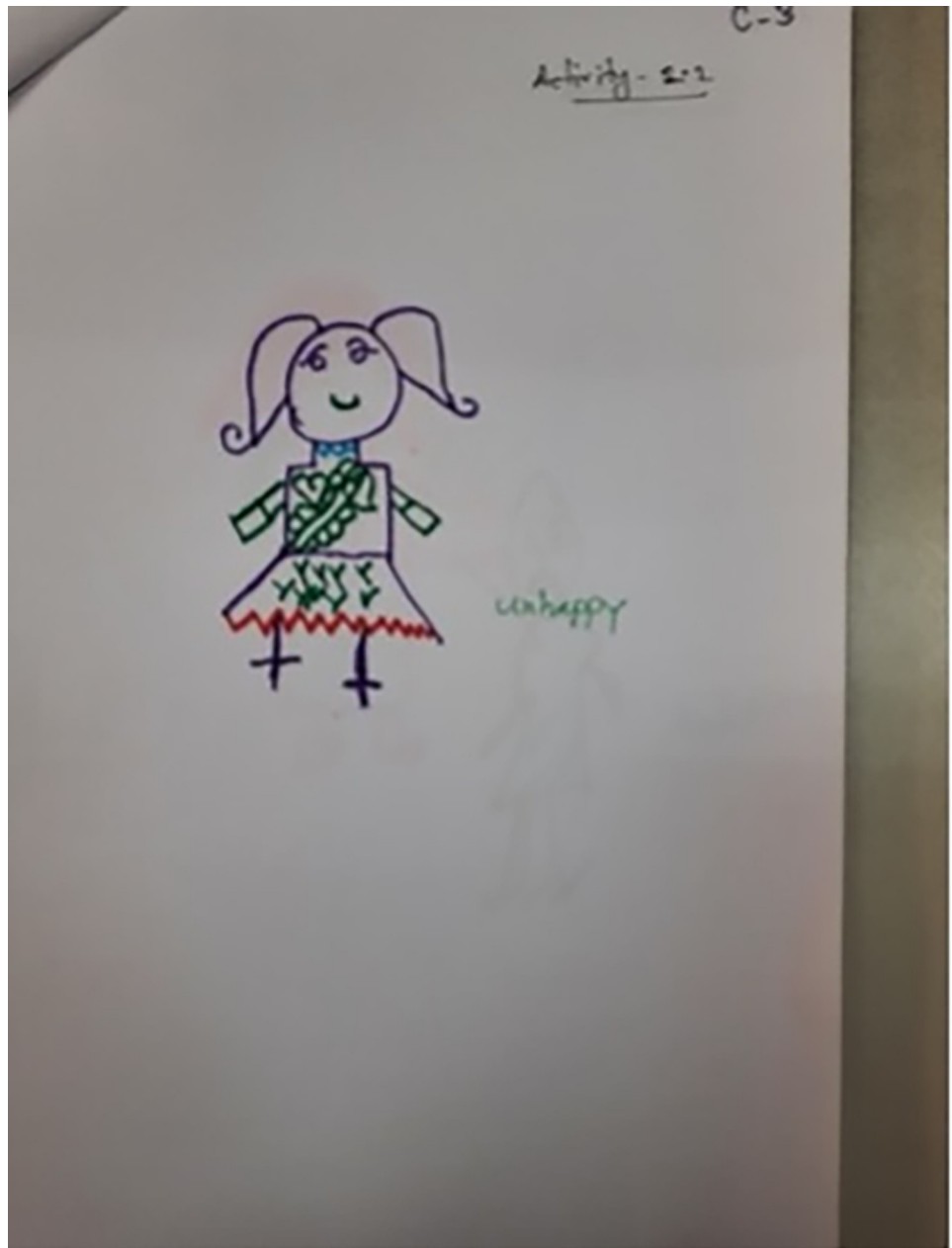

**Fig 3. 'Unhappy' (CB8).** Adding descriptors to explain drawings (Photographs by Sudipta Das Gupta). Note that the descriptions of the emotion do not clearly match the faces of the drawn figures.

An interviewee in Adjumani District noted that '*the ice breaking parts could be emphasised more to add more of the local content*' (ADI1) with ADI2 suggesting that '*maybe just adding an aspect of a song, or a story that you know about that imaginary character [about self-wetting] maybe they would have given us more talking*'.

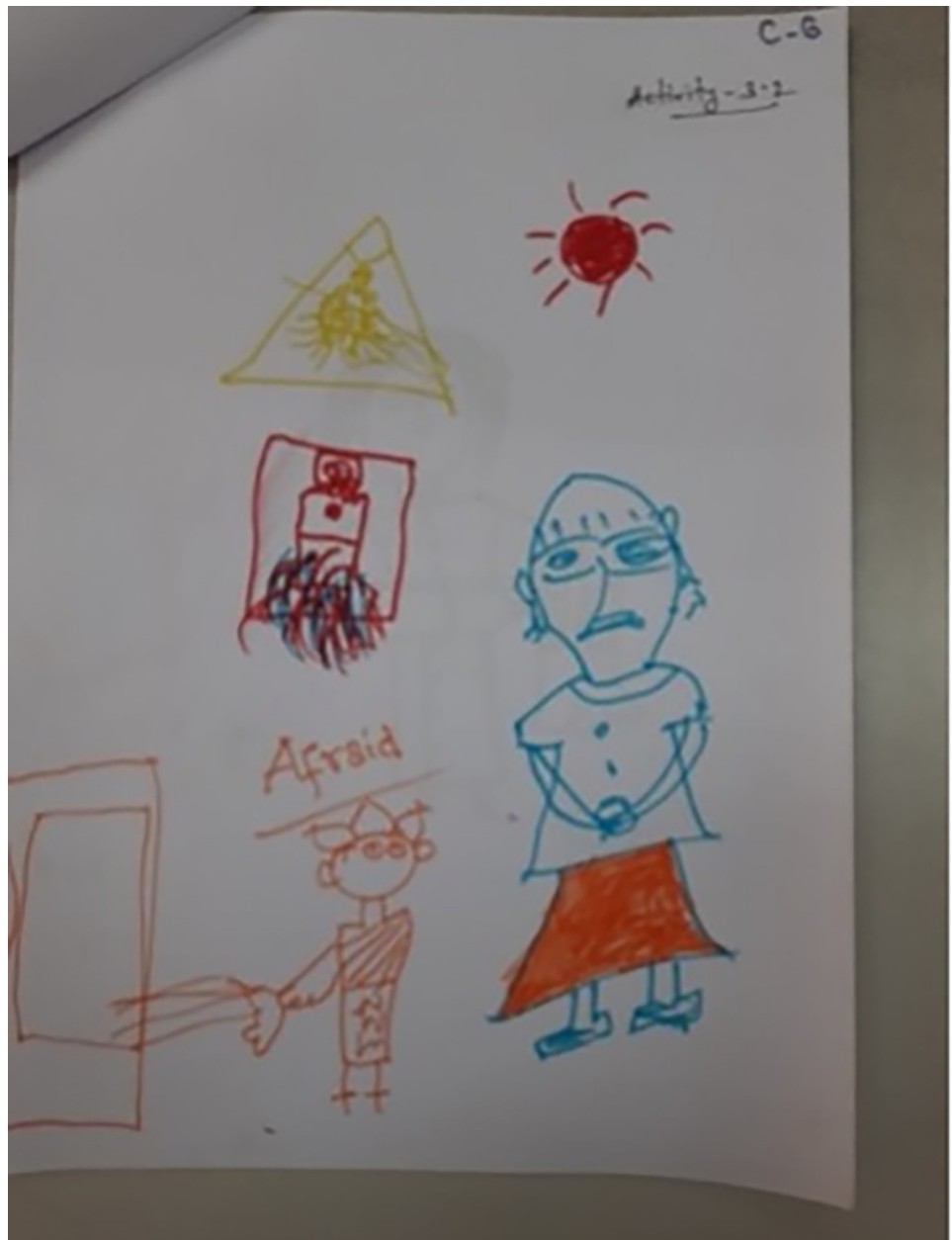

**Fig 4. 'Afraid' (CB8).** Adding descriptors to explain drawings (Photographs by Sudipta Das Gupta). Note that the descriptions of the emotion do not clearly match the faces of the drawn figures.

## Discussion

### Lessons learned

The Story Book methodology was used in two very different contexts–refugee settlements in Adjumani District, Uganda and refugee camps in Cox's Bazar, Bangladesh–but strikingly similar results were found. Participants in both settings showed an awareness that children do wet themselves, and drawings to demonstrate the consequences of this for the child and their caregiver / teacher / friends were largely expressing significantly negative actions and/or emotions

(forthcoming manuscripts will provide further details of the specific findings in each context). The Story Book methodology therefore proved–in Adjumani District and Cox's Bazar at least– that self-wetting is a public health challenge that needs to be on the agenda of humanitarian practitioners. However, this evaluation must conclude that whilst with some amendments the Story Book methodology could be the research tool of choice to prompt discussions with older children on personal and highly sensitive issues, it is unlikely that it could ever be a suitable methodology to be used in a humanitarian setting, or at least in the immediate onset of a crisis.

### Reflections on the Story Book methodology as a research tool for a humanitarian context

To be successful in a humanitarian context, a research methodology must be flexible, adaptive and iterative; produce quick, real-time data; and deeply engage with the affected population to enable trust, improve research design and facilitate the dissemination of findings [7–9]. The results demonstrate that the Story Book methodology fails to meet these requirements largely because it is so resource-intensive (practicality): context-specific adaptations cannot be done quickly (adaptation); it is dependent on the recruitment and training of data collectors familiar with the local culture and fluent in the languages spoken by participants (practicality); and sufficient engagement with the affected population to build trust with both the participants and wider community takes weeks rather than days (practicality). Further, the data collectors struggled to interpret the data and even then could only hypothesise practical recommendations to improve the wellbeing of displaced children that self-wet (implementation).

Given the resources (financial, technical, human and time) needed to prepare for, conduct and analyse the sessions, the Story Book methodology is therefore better suited as an occasional research tool to determine general needs (most likely at a later stage of an emergency response) rather than as part of an initial needs assessment in an unstable and disrupted environment [27, 28].

### Reflections on the Story Book methodology as a research tool to facilitate having conversations with children about self-wetting

The RT approached the development of the Story Book methodology from an initial stance of 'involving children in research is the right thing to do' and concluded that children should participate because the matter being researched concerned them directly; the RT had the capacity to conduct the research and act on the findings; and the research could be conducted ethically (using the Story Book methodology) (S1 File and 16).

This evaluation has found that the Story Book methodology was not always implemented as designed (not all activities always took place, most sessions ran over the intended maximum time of 90 minutes, and participants became tired and lost concentration at times); and not all children found the methodology acceptable (they struggled to provide answers to the questions being asked and to draw a response). Yet the sessions did provide a safe space for children to do a creative activity that they may not usually do on a daily basis; there were no indications that any child found the sessions to be traumatic or harmful; and the study provided an opportunity–probably for the first time–for children to have their voices heard on a personal and highly sensitive health issue (migrant research to date has tended to prioritise adult frames of reference on children's health-related experiences and needs even though adults do not necessarily make good proxies for children [4, 29]).

Recommendations (S4 File) have been made to improve the implementation and acceptability of the Story Book methodology. If the suggested changes are made, we/the authors believe that the revised Story Book methodology would be a suitable tool to prompt discussions with

children aged five to eleven on personal and highly sensitive issues. Where time allows, the methodology could be further adapted to be integrated into a series of activities implemented by teachers (when they have an established and trusting relationship with both the children and caregivers) over several days (or longer). This assumes that the children would be comfortable completing the activities in a school setting, with their teachers, and with known peers, and of course this may not hold true for all participants due to, for example, existing power imbalances between adults and children and/or a shyness to participate around friends [30, 31].

## Limitations

This research project took place during the COVID-19 outbreak. It had been planned that RT members who developed the Story Book methodology (CRS, JR, DJB) would travel to Bangladesh and Uganda to work directly with PIU and WVB to contextualise the methodology before deployment and adapt it as necessary during data collection. Due to travel restrictions, CRS, JR and DJB were unable to visit Bangladesh or Uganda during the lifetime of the project, so this contextualisation had to take place remotely. After contextualisation, because travel restrictions were still in place, MUA and EW, who were collaborating with the RT on other projects, were invited to partner on this work, training local data collectors and overseeing data collection. However, they had to be trained remotely in the Story Book methodology by CRS and DJB. Due to internet speeds/services and unreliable electricity in Adjumani and Cox's Bazar, online contextualisation and training was difficult and often disjointed. It is likely that if face to face contextualisation and training had been possible, some of the pitfalls of the Story Book approach would have been identified and rectified before implementation.

The number of interviews that could be conducted with adults known to have participated in the planning and/or facilitation of the Story Book sessions was limited by the lead author only being able to conduct research in English. This was mitigated to some degree by the facilitators providing fieldnotes in their preferred language, which were translated to English for analysis.

## Conclusion

This paper has evaluated the Story Book methodology as a means to facilitate having conversations with displaced children on highly sensitive topics to inform humanitarian programming. The Story Book sessions held in Adjumani District and Cox's Bazar using the methodology demonstrated that children are aware about self-wetting and tend to associate it with significantly negative emotions and consequences. This justifies considering self-wetting as a public health challenge. However, the Story Book methodology wasn't implemented as designed; it is not easily adapted to, or practical to implement in, humanitarian settings; and it was not acceptable to all participants and facilitators as a research tool to prompt discussions about self-wetting.

Changes have been recommended to improve how the Story Book methodology is implemented in practice and accepted by participants. With such changes it could yet be useful as a research tool to better understand the general needs of children experiencing self-wetting. However, given how resource-intensive the Story Book methodology is, it is unlikely that it could ever be a suitable research tool to be used in a humanitarian setting, or at least in the immediate onset of a crisis.

## Supporting information

**S1 File. Extracts from the PhD thesis of C. rosato-scott detailing development of the Story Book methodology.**
(DOCX)

**S2 File. Data on sessions held in Cox's Bazar.** Table A. Story Book sessions held in Cox's Bazar. Table B. Answers given* on how the Hero feels after self-wetting. Table C. Answers given* on reactions of others to the Hero after self-wetting. Table D. Answers given* when asked why the Hero wet the bed. Table E. Answers given* when asked how the Hero could stop wetting the bed.
(DOCX)

**S3 File. Inclusivity in global research checklist.**
(DOCX)

**S4 File. Recommendations.** Table A. Changes proposed to the Story Book session agenda.
(DOCX)

## Acknowledgments

The RT would like to acknowledge the many individuals and organisations who contributed to the completion of this project, particularly research trainers and data collectors, administrative staff at all organisations, ethical application reviewers and the project's Advisory Board. Most importantly they would like to thank the children who participated in the Story Book sessions.

## Author Contributions

**Conceptualization:** Claire Rosato-Scott, Barbara E. Evans, Joanne Rose, Dani J. Barrington.

**Data curation:** Claire Rosato-Scott.

**Formal analysis:** Claire Rosato-Scott.

**Funding acquisition:** Claire Rosato-Scott, Barbara E. Evans, Joanne Rose, Dani J. Barrington.

**Investigation:** Claire Rosato-Scott.

**Methodology:** Claire Rosato-Scott, Barbara E. Evans, Joanne Rose, Dani J. Barrington.

**Project administration:** Barbara E. Evans, Joanne Rose, Dani J. Barrington.

**Supervision:** Barbara E. Evans, Joanne Rose, Dani J. Barrington.

**Validation:** Claire Rosato-Scott.

**Writing – original draft:** Claire Rosato-Scott.

**Writing – review & editing:** Claire Rosato-Scott, Mahbub-Ul Alam, Barbara E. Evans, Joanne Rose, Eleanor Wozei, Dani J. Barrington.

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
