## [Decision Letter · Decision Letter 0]

18 Nov 2022

PGPH-D-22-01395

Understanding children’s experiences of self-wetting in humanitarian contexts: An evaluation of the Story Book methodology

Dear Dr. Barrington,

Thank you for submitting your manuscript to PLOS Global Public Health. After careful consideration, we feel that it has merit but does not fully meet PLOS Global Public Health’s publication criteria as it currently stands. Therefore, we invite you to submit a revised version of the manuscript that addresses the points raised during the review process.

The manuscript has been evaluated by two reviewers, and their comments are available below.

Although both reviewers are positive about the manuscript, they have raised a number of concerns that need attention. In particular, they request additional information on methodological aspects of the study.

Could you please revise the manuscript to carefully address the concerns raised?

We look forward to receiving your revised manuscript.

Kind regards,

Steve Zimmerman, PhD

PLOS Staff Editor

Journal Requirements:

2. Please send a completed 'Competing Interests' statement, including any COIs declared by your co-authors. If you have no competing interests to declare, please state "The authors have declared that no competing interests exist". Otherwise please declare all competing interests beginning with the statement "I have read the journal's policy and the authors of this manuscript have the following competing interests:"

3. Please amend your detailed Financial Disclosure statement. This is published with the article. It must therefore be completed in full sentences and contain the exact wording you wish to be published.

b. If any authors received a salary from any of your funders, please state which authors and which funders.

4. Please provide separate figure files in .tif or .eps format only and remove any figures embedded in your manuscript file. Please also ensure that all files are under our size limit of 10MB.

Additional Editor Comments (if provided):

Reviewers' comments:

Reviewer's Responses to Questions

**Comments to the Author**

1. Does this manuscript meet PLOS Global Public Health’s publication criteria? Is the manuscript technically sound, and do the data support the conclusions? The manuscript must describe methodologically and ethically rigorous research with conclusions that are appropriately drawn based on the data presented.

Reviewer #1: Yes

Reviewer #2: Partly

2. Has the statistical analysis been performed appropriately and rigorously?

Reviewer #1: N/A

Reviewer #2: Yes

3. Have the authors made all data underlying the findings in their manuscript fully available (please refer to the Data Availability Statement at the start of the manuscript PDF file)?

Reviewer #1: Yes

Reviewer #2: Yes

4. Is the manuscript presented in an intelligible fashion and written in standard English?

Reviewer #1: Yes

Reviewer #2: Yes

5. Review Comments to the Author

Reviewer #1: The paper is interesting, well written, and addresses an underinvestigated issue. Even if it emerges across the paper, I reccommend to state more clearly why self-wetting was chosen as a problem to be investigated: consequences are described, but what are its dimensions, occurrence, why it is worthy to be investigated in that context?

Also it will be beneficial to describe more why the Story Book was chosen to be tested as methodology and not another methodology. Due to the fact that the Story Book is not the best methodology to address this problem in humanitarian settings, which methods can be appropriate? or what about next steps to identify more appropriate tools for such contexts?

Reviewer #2: The paper purpose is highly important and is recommended to be further improved especially for sensitive topics.

However, because this is a methodological paper, it is important that the methodology part of the paper needs to be more detailed in order for the recommendations and conclusion made in the paper to be justified and understandable .

1. The setting; (place) in which the sessions were conducted with their children. Was it in their homes, schools, or unfamilar places, were the children in each session familiar with each other before the sessions, were the parents( or other caregivers) of the children around during the sessions etc. All these factors may affect children's response during the sessions.

2. The methodology needs to include the description of the facilitators of the sessions. How were they selected? This should include their gender, educational level, experience in research with children, their previous level of interaction or familiarity with the children and the community.

3. Were local languages used with the children

4. What is the socioeconomic status or educational level of the children involved in each session, Do children in each session have similar levels of socioeconomic or educational status.

5 What is the total number of children participants. Descriptive statistics ( gender, age) of the participant should be included.

5. Asides the table 2 that shows that sessions were done in age groups. It seems that two age groups were used from the table ; the text should also indicate how children were grouped for the sessions.

6. Do the children differ in participation level during the sessions based on their age groups. Please include this in the result sessions

6. PLOS authors have the option to publish the peer review history of their article (what does this mean?). If published, this will include your full peer review and any attached files.

**Do you want your identity to be public for this peer review?** For information about this choice, including consent withdrawal, please see our Privacy Policy.

Reviewer #1: No

Reviewer #2: No

---

## [Decision Letter · Decision Letter 1]

13 Apr 2023

Understanding children’s experiences of self-wetting in humanitarian contexts: An evaluation of the Story Book methodology

PGPH-D-22-01395R1

Dear Dr Barrington,

We are pleased to inform you that your manuscript 'Understanding children’s experiences of self-wetting in humanitarian contexts: An evaluation of the Story Book methodology' has been provisionally accepted for publication in PLOS Global Public Health.

Best regards,

Julia Robinson

Executive Editor

Reviewer Comments (if any, and for reference):

Reviewer's Responses to Questions

**Comments to the Author**

1. If the authors have adequately addressed your comments raised in a previous round of review and you feel that this manuscript is now acceptable for publication, you may indicate that here to bypass the “Comments to the Author” section, enter your conflict of interest statement in the “Confidential to Editor” section, and submit your "Accept" recommendation.

Reviewer #1: All comments have been addressed

2. Does this manuscript meet PLOS Global Public Health’s publication criteria? Is the manuscript technically sound, and do the data support the conclusions? The manuscript must describe methodologically and ethically rigorous research with conclusions that are appropriately drawn based on the data presented.

Reviewer #1: Yes

3. Has the statistical analysis been performed appropriately and rigorously?

Reviewer #1: N/A

4. Have the authors made all data underlying the findings in their manuscript fully available (please refer to the Data Availability Statement at the start of the manuscript PDF file)?

Reviewer #1: Yes

5. Is the manuscript presented in an intelligible fashion and written in standard English?

Reviewer #1: Yes

6. Review Comments to the Author

Reviewer #1: (No Response)

7. PLOS authors have the option to publish the peer review history of their article (what does this mean?). If published, this will include your full peer review and any attached files.

**Do you want your identity to be public for this peer review?** For information about this choice, including consent withdrawal, please see our Privacy Policy.

Reviewer #1: No
